# Cytoprotective effects of (E)-N-(2-(3, 5-dimethoxystyryl) phenyl) furan-2-carboxamide (BK3C231) against 4-nitroquinoline 1-oxide-induced damage in CCD-18Co human colon fibroblast cells

Huan Huan Tan[1], Noel Francis Thomas[2], Salmaan Hussain Inayat-Hussain[3,4], Kok Meng Chan[1]*

1 Center for Toxicology and Health Risk Studies, Faculty of Health Sciences, Universiti Kebangsaan Malaysia, Kuala Lumpur, Malaysia, 2 Methodist College Kuala Lumpur, Kuala Lumpur, Malaysia, 3 Product Stewardship and Toxicology, Group Health, Safety, Security and Environment, Petroliam Nasional Berhad (PETRONAS), Kuala Lumpur, Malaysia, 4 Department of Environmental Health Sciences, Yale School of Public Health, New Haven, CT, United States of America

* chan@ukm.edu.my

**Data Availability Statement:** All relevant data are within the manuscript.

## Abstract

Stilbenes are a group of chemicals characterized with the presence of 1,2-diphenylethylene. Previously, our group has demonstrated that synthesized (E)-N-(2-(3, 5-dimethoxystyryl) phenyl) furan-2-carboxamide (BK3C231) possesses potential chemopreventive activity specifically inducing NAD(P)H:quinone oxidoreductase 1 (NQO1) protein expression and activity. In this study, the cytoprotective effects of BK3C231 on cellular DNA and mitochondria were investigated in normal human colon fibroblast, CCD-18Co cells. The cells were pretreated with BK3C231 prior to exposure to the carcinogen 4-nitroquinoline 1-oxide (4NQO). BK3C231 was able to inhibit 4NQO-induced cytotoxicity. Cells treated with 4NQO alone caused high level of DNA and mitochondrial damages. However, pretreatment with BK3C231 protected against these damages by reducing DNA strand breaks and micronucleus formation as well as decreasing losses of mitochondrial membrane potential ($\Delta\Psi m$) and cardiolipin. Interestingly, our study has demonstrated that nitrosative stress instead of oxidative stress was involved in 4NQO-induced DNA and mitochondrial damages. Inhibition of 4NQO-induced nitrosative stress by BK3C231 was observed through a decrease in nitric oxide (NO) level and an increase in glutathione (GSH) level. These new findings elucidate the cytoprotective potential of BK3C231 in human colon fibroblast CCD-18Co cell model which warrants further investigation into its chemopreventive role.

## 1. Introduction

Cancer-related mortality has increased tremendously and is expected to further increase despite emerging medical improvements [1]. The global incidence of cancer is estimated to

**Funding:** This research was financially supported by Exploratory Research Grant Scheme (ERGS) from Kementerian Pendidikan Malaysia (Grant number: ERGS/1/2013/SKK03/UKM/02/1, URL: http://jpt.mohe.gov.my/portal/penyelidikan/mygrants) and Dana Impak Perdana (DIP) from Universiti Kebangsaan Malaysia (Grant number: DIP-2012-024, URL: http://research.ukm.my/) The funders had no role in study design, data collection and analysis, decision to publish, or preparation of the manuscript.

**Competing interests:** The authors have declared that no competing interests exist.

have risen in 2018 with colorectal cancer being the third most commonly diagnosed cancer and is ranked second in terms of mortality due to poor prognosis worldwide [2]. In Malaysia, cancer is the third most common cause of death after cardiovascular diseases and respiratory diseases. According to Malaysia National Cancer Registry (MNCR) Report 2007–2011, colorectal cancer is the second most common cancer [3].

Advances in costly surgical and medical therapies for primary and metastatic colorectal cancer have had limited impact on cure rates and long-term survival [4,5]. The shift of momentum towards chemoprevention is the result of several observations. These include local recurrences after surgery, treatment-induced long-term complications, chemotherapy-induced adverse effects, as resistance to chemotherapy due to multidrug resistance phenotypes and tumour heterogeneity [6–8].

Chemopreventive approaches have effectively decreased cancer incidence rates such as for lung cancer and cervical cancer [2]. The cancer chemoprevention approach exploits nontoxic natural or synthetic pharmacological agents to prevent, block or reverse the multistep processes of carcinogenesis [9,10]. Chemopreventive agents inhibit the invasive development of cancer by affecting the three defined stages of carcinogenesis namely initiation, promotion and progression which are induced by carcinogens through genetic and epigenetic mechanisms [11,12].

Exposure of cells to carcinogens causes DNA mutation and leads to accumulation of additional genetic changes through sustained cell proliferation. This rapid and irreversible process is known as tumour initiation, the first stage of carcinogenesis. Tumour promotion, which is referred to as the lengthy and reversible second stage of carcinogenesis, involves the selective clonal expansion of initiated cells to produce preneoplastic lesions which enables additional mutations to accumulate. The final stage of carcinogenesis, tumour progression, involves neoplastic transformation after accumulating chromosomal aberrations and karyotypic instability resulting in metastatic malignancy [13,14].

Altered cellular redox status and disrupted oxidative homeostasis play key roles in cancer development by enhancing DNA damage and modifying key cellular processes such as cell proliferation and apoptosis [15]. Oxidative/nitrosative stress is the result of disequilibrium between reactive oxygen species (ROS)/reactive nitrogen species (RNS) and antioxidants [16]. If oxidative/nitrosative stress persists, this may lead to modification of signal transduction and gene expression, which in turn may lead to mutation, transformation and progression of cancer [17,18].

Stilbenes are produced in the skin, seeds, leaves and sapwood of a wide variety of plant species including dicotyledon angiosperms such as grapevine (*Vitis vinifera*), peanut (*Arachis hypogaea*) and Japanese knotweed (Fallopia Japonica); monocotyledons like sorghum (*Sorghum bicolor*) and gymnosperms such as several *Pinus* and *Picea* species [19–21]. They are a well-known class of naturally occurring phytochemicals acting as antifungal phytoalexins, providing protection against UV light exposure and also involved in bacterial root nodulation and coloration [19,22–24]. These compounds bear the core structure of 1,2-diphenylethylene in which two benzene rings are separated by an ethanyl or ethenyl bridge [25].

Despite being known as plant defense compounds, stilbenes have an enormous diversity of effects on biological and cellular processes applicable to human health, particularly in chemoprevention. Resveratrol, as the biosynthetic precursor of most oligostilbenoids, has been known to possess a myriad of biological activities such as anticancer, antioxidant, anti-aging, antimicrobial, cardioprotection, anti-diabetes, anti-obesity, and anti-inflammation [26–33]. However, low water solubility and poor bioavailability are the major setbacks to the exploitation of these biological activities [34,35].

**Fig 1. Chemical structure of BK3C231 [36].**

Our group has previously demonstrated that synthetic stilbene BK3C231 (Fig 1) potently induced antioxidant gene NQO1 as a detoxifying mechanism in human embryonic hepatocytes, WRL-68 cells [36]. Therefore in this study, we propose to elucidate the cytoprotective effects of BK3C231 using normal human colon fibroblast CCD-18Co cells. We anticipate this study will build a strong base and accelerate the development of BK3C231 as a potential drug for chemoprevention.

## 2. Materials and methods

### 2.1 Test compounds

(E)-N-(2-(3, 5-Dimethoxystyryl) phenyl) furan-2-carboxamide (BK3C231) was synthesized and contributed by Dr. Noel Francis Thomas and Dr. Kee Chin Hui from Department of Chemistry, Faculty of Science, University of Malaya (Kuala Lumpur, Malaysia). 4-Nitroquinoline 1-oxide (4NQO) (Cas. No: 56-57-5, ≥98% purity) was purchased from Sigma-Aldrich (St. Louis, MO, USA). Stock solution of BK3C231 at 100mM and 4NQO at 25mg/mL were prepared by dissolving the compounds in solvent dimethyl sulfoxide (DMSO; Thermo Fisher Scientific, Waltham, MA, USA).

### 2.2 Cell culture

The normal human colon fibroblast CCD-18Co cell line (ATCC CRL-1459) was obtained from the American Type Culture Collection (ATCC; Manassas, VA, USA). CCD-18Co cells were grown in Minimum Essential Medium (MEM; Gibco, Grand Island, NY, USA) supplemented with 10% fetal bovine serum (FBS; Biowest, Nuaillé, France) and 1% 100x Antibiotic-

Antimycotic solution (Nacalai Tesque, Kyoto, Japan). All cells were between passages 3–5 for all experiments and maintained at 37˚C with 5% $CO_2$.

## 2.3 MTT cytotoxicity assay

CCD-18Co cells were seeded in 96-well microplate (Nest Biotechnology, Jiangsu, China) at the concentration of 5 x $10^4$ cells/mL in a volume of 200 μL per well. The seeded cells were incubated under 5% $CO_2$ at 37˚C for 24 hours prior to respective compound treatments at different timepoints. After incubation, 20 μL of Thiazolyl Blue Tetrazolium Bromide (MTT; Sigma-Aldrich, St. Louis, MO, USA) solution (5mg/mL in PBS) was added to the treated cells and further incubated for 4 hours at 37˚C. Subsequently, the total medium in each well was discarded and the crystalline formazan was solubilised using 200 μL DMSO. For complete dissolution, the plate was incubated for 15 minutes followed with gentle shaking for 5 minutes. The cytotoxicity of BK3C231 and 4NQO was assessed by measuring the absorbance of each well at 570 nm using iMark™ microplate reader (Bio-Rad Laboratories, Hercules, CA, USA). Mean absorbance for each compound concentration was expressed as a percentage of vehicle control absorbance and plotted versus compound concentration. The inhibitory concentration that kills 50% of cell population ($IC_{50}$) represents the compound concentration that reduced the mean absorbance at 570 nm to 50% of those in the vehicle control wells. [dx.doi.org/10.17504/protocols.io.bdp6i5re]

## 2.4 Alkaline comet assay

Seeded cells (5 x $10^4$ cells/mL) in 6-well plate (Nest Biotechnology, Jiangsu, China) were pretreated with BK3C231 at 6.25 μM, 12.5 μM, 25 μM and 50 μM for 2 hours prior to 4NQO treatment at 1 μM for 1 hour. Following incubation, detached cells in the medium were collected and added back to trypsinised cells. Then, the cell suspension was transferred to the tube for centrifugation (450 x g/5 minutes at 4˚C). The supernatant was removed and pellet was washed with $Ca^{2+}$- and $Mg^{2+}$-free PBS and re-centrifuged. The pellets left at the bottom were mixed thoroughly with 80 μl of 0.6% w/v LMA (Sigma-Aldrich, St. Louis, MO, USA). The mixture was then pipetted onto the hardened 0.6% w/v NMA (Sigma-Aldrich, St. Louis, MO, USA) as the first layer gel on the slide. Cover slips were placed to spread the mixture and slides were left on ice for LMA to solidify. Following removal of the cover slips, the embedded cells were lysed in a lysis buffer containing 2.5M NaCl (Merck Milipore, Burlington, MA, USA), 1 mM $Na_2$EDTA (Sigma-Aldrich, St. Louis, MO, USA), 10 mM Tris (Bio-Rad Laboratories, Hercules, CA, USA) and 1% Triton X-100 (Sigma-Aldrich, St. Louis, MO, USA) overnight at 4˚C. After lysis, the slides were soaked in electrophoresis buffer solution for 20 minutes for DNA unwinding before electrophoresis at 300 mA, 25V for 20 minutes. Subsequently, the slides were rinsed with neutralising buffer for 5 minutes and stained with 30 μL of 50 μg/mL ethidium bromide (EtBr; Sigma-Aldrich, St. Louis, MO, USA) solution. Slides were left overnight at 4˚C before analyzing with Olympus BX51 fluorescence microscope (Tokyo, Japan) equipped with 590 nm filter. DNA damage scoring was performed on 50 cells per slide whereby tail moment representing the product of tail length and fraction of total DNA in tail was quantified using Comet Score™ software (TriTek Corp, Sumerduck, VA, USA). [dx.doi.org/10.17504/protocols.io.bdqgi5tw]

## 2.5 Cytokinesis-block micronucleus (CBMN) assay

Seeded cells (5 x $10^4$ cells/mL) in 6-well plate were pretreated with BK3C231 at 6.25 μM, 12.5 μM, 25 μM and 50 μM for 2 hours prior to 4NQO treatment at 1 μM for 2 hours. After incubation, cells were treated with 4.5 μg/mL Cytochalasin B (Sigma-Aldrich, St. Louis, MO,

USA) for 24 hours to block cytokinesis. The cells were then harvested and centrifuged (450 x g/5 minutes at 4˚C). The supernatant was removed and pellet was resuspended with 300 μL of 0.075M KCl solution for 5 minutes. The cells were then fixed with Carnoy's solution consisting of acetic acid (Sigma-Aldrich, St. Louis, MO, USA) and methanol (HmbG Chemicals, Hamburg, Germany) prepared at the ratio of 1:3 and spreaded on glass slides which were placed on a slide warmer. The slides were dried overnight and stained with 30 μL of 20 μg/mL acridine orange (AO; Sigma-Aldrich, St. Louis, MO, USA) prior to fluorescence microscopic observation. The number of viable mononucleated, binucleated and multinucleated cells per 500 cells were scored to derive Nuclear Division Index (NDI) and frequency of micronucleus in 1,000 binucleated cells was measured. [dx.doi.org/10.17504/protocols.io.bdqhi5t6]

## 2.6 Mitochondrial membrane potential (ΔΨm), mitochondrial mass and ROS assessment

The treated cells (5 x $10^4$ cells/mL) were collected by centrifugation (450 x g/5 minutes at 4˚C). The supernatant was discarded and pellet was resuspended with 1 mL fresh prewarmed FBS-free MEM with addition of 1 μL of 50 μM tetramethylrhodamine ethyl ester (TMRE; Thermo Fisher Scientific, Waltham, MA, USA), 5 mM nonyl acridine orange (NAO; Sigma-Aldrich, St. Louis, MO, USA), 10 mM hydroethidine (HE; Thermo Fisher Scientific, Waltham, MA, USA) or 10 mM 2',7'-dichlorodihydrofluorescein diacetate (DCFH-DA; Thermo Fisher Scientific, Waltham, MA, USA). The cells stained with TMRE or NAO were incubated for 15 minutes at 37˚C whereas cells stained with HE and DCFH-DA were incubated for 30 minutes at 37˚C in the dark. After incubation, the cells were centrifuged (450 x g/5 minutes at 4˚C) and pellet was washed with 1 mL chilled PBS solution. The supernatant was discarded and 500 μL of chilled PBS was used to resuspend the pellets. The stained cell suspension was transferred to flow tubes and analyzed using FACSCanto II Flow Cytometer (BD Biosciences, San Jose, CA, USA). [dx.doi.org/10.17504/protocols.io.bdqii5ue; dx.doi.org/10.17504/protocols.io.bdqji5un; dx.doi.org/10.17504/protocols.io.bdqki5uw; dx.doi.org/10.17504/protocols.io.bdqmi5u6]

## 2.7 Intracellular Nitric Oxide (NO) assessment using BD Pharmingen™ orange nitric oxide (NO) probe staining

The seeded cells (5 x $10^4$ cells/mL) were pre-stained with 1 μL Orange NO probe (BD Biosciences, San Jose, CA, USA) per 500 μL cell suspension for 30 minutes. The cells were then pretreated with BK3C231 at 50 μM and positive control, resveratrol (Sigma-Aldrich, St. Louis, MO, USA) at 25 μM for 2 hours, 4 hours, 6 hours, 12 hours and 24 hours prior to 4NQO treatment at 1 μM for 1 hour. The stained and treated cells were centrifuged (450 x g/5 minutes at 4˚C) and pellet was washed with 1 mL chilled PBS solution. The supernatant was discarded and 500 μL of chilled PBS was used to resuspend the pellets. The stained cell suspension was transferred to flow tubes and analyzed using FACSCanto II Flow Cytometer (BD Biosciences, San Jose, CA, USA). [dx.doi.org/10.17504/protocols.io.bdqni5ve]

## 2.8 Extracellular Nitric Oxide (NO) assessment using griess reagent

CCD-18Co cells were seeded in culture dish (60 x 15 mm) at the concentration of 5 x $10^4$ cells/ mL. The seeded cells were incubated under 5% $CO_2$ at 37˚C for 24 hours. The cells were then pretreated with BK3C231 at 50 μM and positive control, resveratrol at 25 μM for 2 hours, 4 hours, 6 hours, 12 hours and 24 hours prior to 4NQO treatment at 1 μM for 1 hour. Subsequently, 100 μL of culture medium from each sample was collected and mixed with the same volume of Griess reagent (1% sulfanilamide in 5% phosphoric acid and 0.1% N-(1-naphthyl)

ethylenediamine (NNED) hydrochloride in distilled water, Merck Milipore, Burlington, MA, USA) in 96-well microplate. Absorbance of the mixture in each well was determined at 570 nm using iMark$^{TM}$ microplate reader (Bio-Rad Laboratories, Hercules, CA, USA). The concentration of nitrite accumulated in the culture was determined in comparison to the sodium nitrite standards. [dx.doi.org/10.17504/protocols.io.bdqpi5vn]

### 2.9 Glutathione (GSH) assessment using ellman's reagent

The treated cells ($5 \times 10^4$ cells/mL) were detached, collected and centrifuged (450 x g/5 minutes at 4˚C). The supernatant was discarded and pellet was resuspended in 100 μL ice-cold lysis buffer (50 mM $K_2HPO_4$, 1 mM EDTA, pH 6.5 and 0.1% v/v Triton X-100, Sigma-Aldrich, St. Louis, MO, USA). The cells were incubated on ice for 15 minutes with gentle tapping from time to time. The crude lysates were cleared by centrifugation (10000 x g/15 minutes at 4˚C). At this point, the lysates were used immediately or stored at -80˚C for a day or two. Then, 50 μl of lysates and GSH standards (two fold dilution from 1.25 mM to 0 mM dissolved in reaction buffer consisting of 0.1 M $Na_2HPO_4.7H_2O$ and 1 mM EDTA, pH 6.5, Sigma-Aldrich, St. Louis, MO, USA) were pipetted into designated wells in a 96-well microplate. After adding 40 μl of reaction buffer (0.1 M $Na_2HPO_4.7H_2O$ and 1 mM EDTA, pH 8), 10 μl of 4 mg/ml 5,5′-dithiobis(2-nitrobenzoic acid) (DTNB; Sigma-Aldrich, St. Louis, MO, USA) in reaction buffer pH 8 was added to wells containing samples and standards. The plate was incubated for 15 minutes at 37˚C. Absorbance of each well was measured at 405 nm using iMark$^{TM}$ microplate reader (Bio-Rad Laboratories, Hercules, CA, USA). The concentration of free thiols in samples was calculated based on GSH standard and expressed as nmol/mg protein after protein concentration was quantified using Bradford's method. [dx.doi.org/10.17504/protocols.io.bdqqi5vw]

### 2.10 Statistical analysis

The data are expressed as the mean ± standard error of mean (S.E.M.) from at least three independent experiments. The statistical significance was evaluated using one-way ANOVA with the Tukey post hoc test used to assess the significance of differences between multiple treatment groups. Differences were considered statistically significant with a probability level of $p < 0.05$.

## 3. Results

### 3.1 Cytotoxic assessment of BK3C231 and 4NQO

The non-cytotoxic concentrations of BK3C231 and 4NQO were determined using MTT cytotoxicity assay. BK3C231 did not show evidence of cytotoxicity up to 50 μM treatment, however an $IC_{50}$ value of 99 μM was observed (Fig 2A). Therefore a series of BK3C231 concentrations ranging from 6.25 μM till 50 μM was used for subsequent experiments. On the other hand, 4NQO treatment exerted no cytotoxicity at 1 hour. However, reduction in cell viability was significant with $IC_{50}$ values observed starting from 2 hours till 24 hours (Fig 2B). Hence, 4NQO concentration at 1 μM was selected to induce genotoxicity and mitochondrial toxicity in subsequent experiments as used by previous studies as well [37,38]. Interestingly, in comparison to 4NQO-treated cells whereby cell viability greatly reduced especially at higher concentrations, BK3C231 was able to suppress 4NQO-induced cytotoxicity by increasing cell viability up to 8-fold with no $IC_{50}$ value observed (Fig 2C).

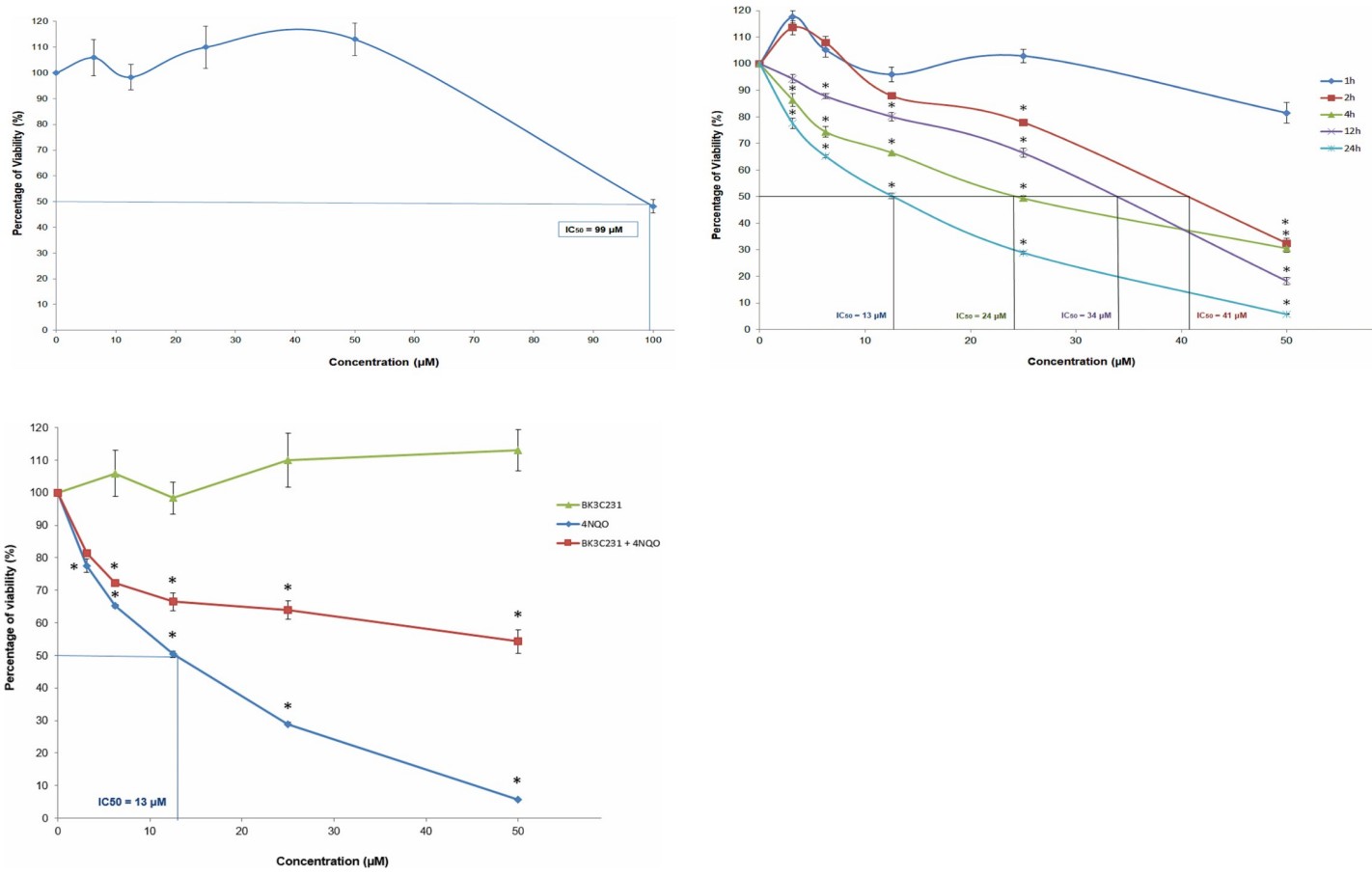

**Fig 2. Effect of BK3C231 and 4NQO on the viability of CCD-18Co cells as assessed by MTT assay.** (A) Cells were treated with BK3C231 from 6.25 μM till 100 μM for 24h. An $IC_{50}$ value of 99 μM was observed. (B) Cells were treated with 4NQO from 3.125 μM till 50 μM for 1h (no $IC_{50}$ value), 2h ($IC_{50}$ value was 41 μM), 4h ($IC_{50}$ value was 24 μM), 12h ($IC_{50}$ value was 34 μM) and 24h ($IC_{50}$ value was 13 μM). (C) Cells were pretreated with BK3C231 at 50 μM for 2h prior to 4NQO treatment from 3.125 μM till 50 μM for subsequent 22h (no $IC_{50}$ value) in comparison to BK3C231-treated (no $IC_{50}$ value) and 4NQO-treated cells ($IC_{50}$ value was 13 μM). Each data point was obtained from three independent experimental replicates and expressed as mean ± SEM of percentage of cell viability. * $p < 0.05$ against negative control.

### 3.2 BK3C231 protection against 4NQO-induced DNA microlesions

Significant DNA damage as indicated by the comet tail which represents DNA strand breaks can be observed in cells treated only with 4NQO. Untreated control cells and BK3C231-treated cells showed intact round nuclear DNA and no DNA strand break was observed at all treated concentrations (Fig 3A). There was also a decrease in comet tail in cells pretreated with BK3C231 when compared with that of cells treated only with 4NQO (Fig 3B). This was further confirmed by quantification of tail moments obtained from comet scoring. Tail moment increased significantly up to 48-fold in 4NQO-treated cells at 28.79 ± 1.02 (p<0.05) over control and BK3C231-treated cells ranging from 0.59 ± 0.11 to 0.68 ± 0.06 (Fig 3C). On the other hand, BK3C231 pretreatment showed a 0.8-fold decrease of 4NQO-induced DNA strand breaks in a concentration-dependent manner, significantly at 50 μM with a tail moment value of 7.21 ± 0.34 (p<0.05) (Fig 3D).

### 3.3 Inhibition of 4NQO-induced DNA macrolesions by BK3C231

The protective role of BK3C231 against 4NQO-induced micronucleus formation was assessed using CBMN assay (Fig 4A). In untreated control cells, a micronucleus frequency level as low

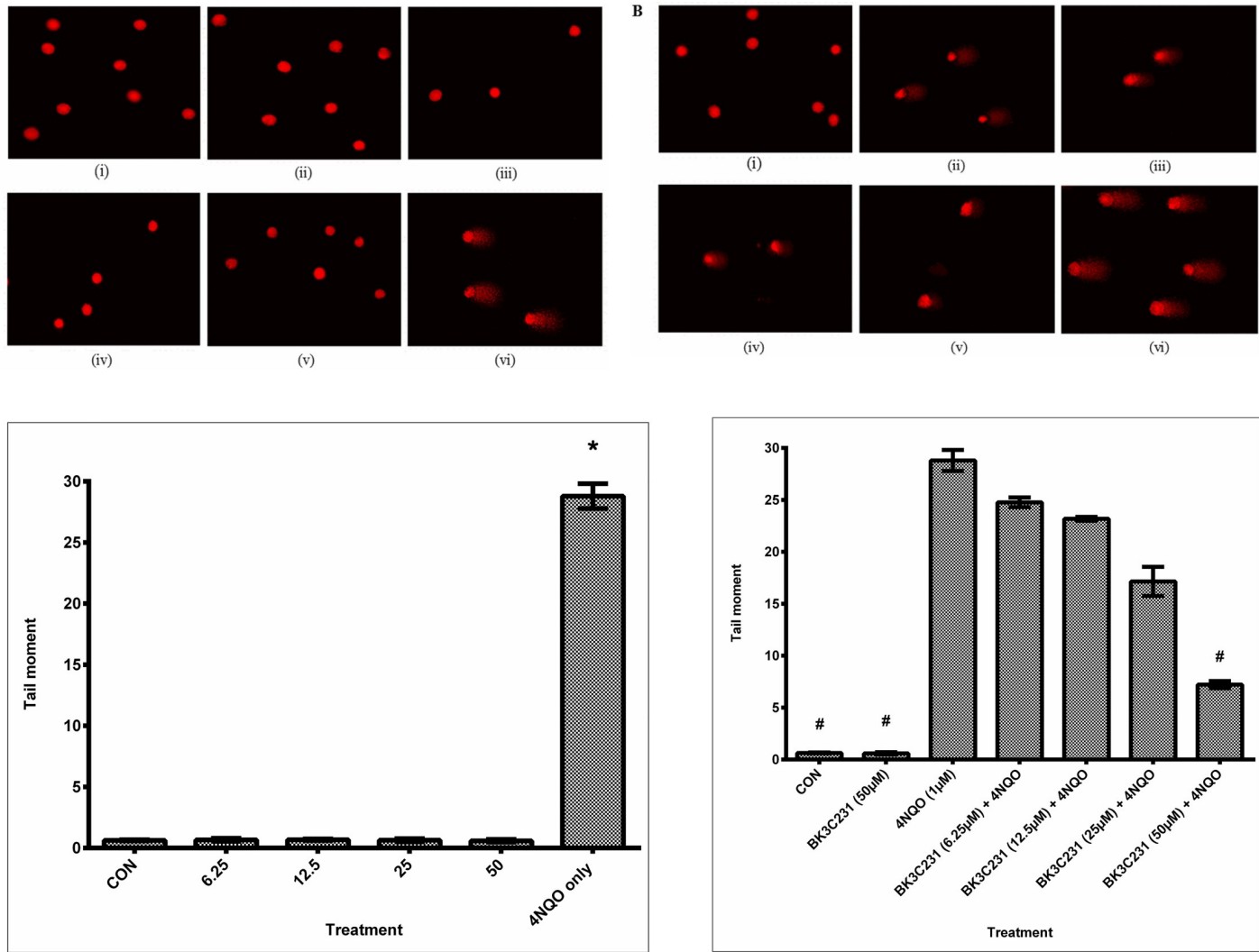

**Fig 3. DNA microlesion assessment in CCD-18Co cells using alkaline comet assay.** (A) Fluorescence microscopic images with EtBr staining of untreated cells (i), cells treated with BK3C231 at 6.25 μM (ii), 12.5 μM (iii), 25 μM (iv) and 50 μM (v) for 24h and cells treated with 4NQO at 1 μM for 1h (vi). (B) Fluorescence microscopic images of untreated cells (i), cells treated with BK3C231 at 6.25 μM (ii), 12.5 μM (iii), 25 μM (iv) and 50 μM (v) for 2h prior to 4NQO induction at 1 μM for 1h and cells treated with 4NQO at 1 μM for 1h (vi). (C) Screening for DNA damage expressed as tail moment in cells treated respectively with BK3C231 from 6.25 μM till 50 μM for 24h and 4NQO at 1 μM for 1h. (D) Cells were pretreated with BK3C231 from 6.25 μM till 50 μM for 2h prior to 4NQO induction at 1 μM for 1h. Each data was obtained from three independent experimental replicates and each data point in (C) and (D) was expressed as mean ± SEM of tail moment. * $p < 0.05$ against negative control, CON (C) and # $p < 0.05$ against positive control, 4NQO only (D).

as 0.23 ± 0.03 was observed. Cells treated with 4NQO significantly demonstrated up to 93-fold increase in frequency of micronucleus in binucleated cells at 21.56 ± 1.36 (p<0.05). However, pretreatment of cells with BK3C231 was shown to cause a maximum of 0.8 fold decrease of 4NQO-induced micronucleus formation in a concentration-dependent manner, significantly at 25 μM with a frequency level of 6.58 ± 0.52 and 50 μM with a frequency level of 3.80 ± 0.47 (p<0.05) (Fig 4B). In addition, the NDI values measured in control, 4NQO-treated cells and BK3C231-treated cells were 1.78 ± 0.01, 1.68 ± 0.03 and 1.79 ± 0.01 respectively. As for cells pretreated with BK3C231 prior to 4NQO induction, the average NDI value measured was 1.72 ± 0.01 (Table 1). All NDI values obtained in this assay indicated normal cell proliferation [39].

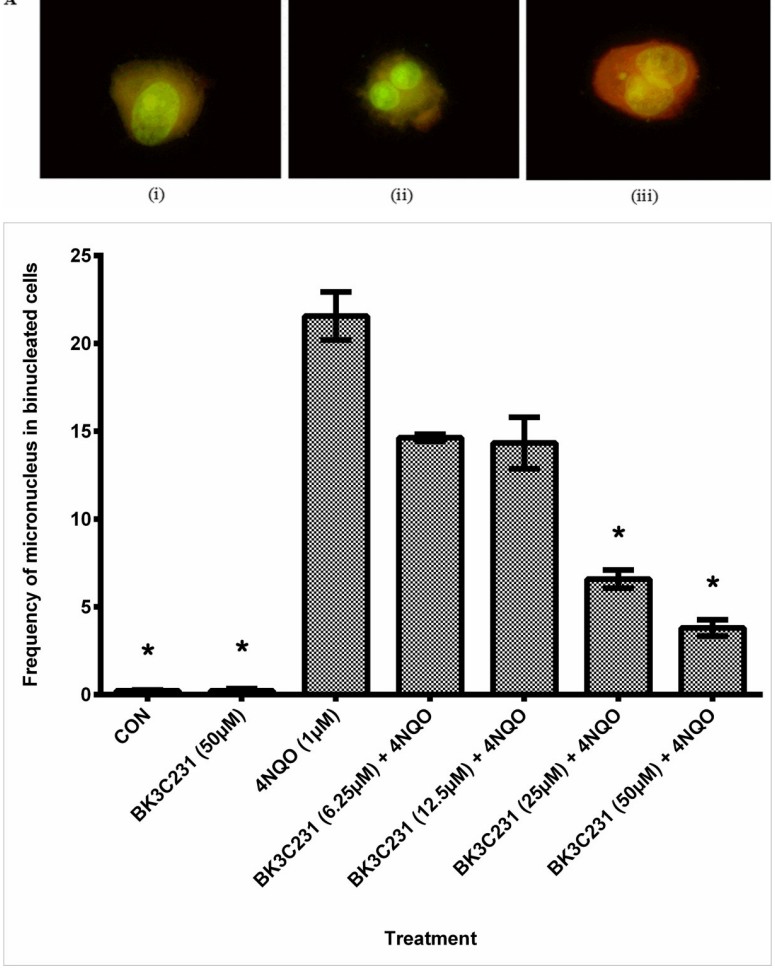

**Fig 4. DNA macrolesion assessment in CCD-18Co cells using CBMN assay.** (A) Fluorescence microscopic images with acridine orange staining of mononucleated cell (i), binucleated cell (ii) and binucleated cell with micronucleus (iii). Cellular nucleus was stained green while cytoplasm was stained orange in this assay. (B) Cells were pretreated with BK3C231 from 6.25 μM till 50 μM for 2h prior to 4NQO induction at 1 μM for 2h. Each data point was obtained from three independent experimental replicates and expressed as mean ± SEM of frequency of micronucleus in binucleated cells. * $p < 0.05$ against positive control, 4NQO only.

**Table 1. NDI values in untreated control cells, BK3C231-treated cells, 4NQO-treated cells and cells pretreated with BK3C231 prior to 4NQO induction.**

| Treatment | NDI value |
|---|---|
| Untreated control | 1.78 ± 0.01 |
| 4NQO 1 μM | 1.68 ± 0.03 |
| BK3C231 50 μM | 1.79 ± 0.01 |
| BK3C231 6.25 μM + 4NQO 1 μM | 1.74 ± 0.01 |
| BK3C231 12.5 μM + 4NQO 1 μM | 1.70 ± 0.01 |
| BK3C231 25 μM + 4NQO 1 μM | 1.74 ± 0.01 |
| BK3C231 50 μM + 4NQO 1 μM | 1.71 ± 0.01 |

NDI, nuclear index division; μM, micromolar; 4NQO, 4-Nitroquinoline 1-oxide.

### 3.4 Cytoprotective role of BK3C231 in 4NQO-induced mitochondrial damages

The cytoprotective role of BK3C231 was further investigated at the mitochondrial level through flow cytometric assessment of $\Delta\Psi$m loss using TMRE staining. Significant loss of $\Delta\Psi$m ($p<0.05$) as indicated by a 1.2-fold increase of TMRE-negative cells from $15.63 \pm 1.09\%$ in control cells to $34.77 \pm 1.29\%$ in 4NQO-treated cells was observed. However, BK3C231 pre-treatment was shown to reduce the amount of TMRE-negative cells significantly to $22.13 \pm 2.51\%$ ($p<0.05$) at 50 μM, thereby protecting the cells against 4NQO-induced $\Delta\Psi$m loss (Fig 5A). Positive control, resveratrol also decreased the amount of TMRE-negative cells significantly to $24.07 \pm 0.22\%$ ($p<0.05$) at 25 μM.

In a bid to further establish the protective role of BK3C231 in mitochondria, cardiolipin level was assessed through flow cytometric analysis using NAO staining. Our study demonstrated significant cardiolipin loss ($p<0.05$) as indicated by a 2.8-fold increase of NAO-negative cells from $5.07 \pm 0.52\%$ in control cells to $19.33 \pm 0.94\%$ in 4NQO-treated cells. However, BK3C231 pretreatment was shown to induce up to 0.4-fold decrease of 4NQO-induced cardiolipin loss in a concentration-dependent manner (Fig 5B). Positive control, resveratrol also reduced the amount of TMRE-negative cells significantly to $12.83 \pm 0.13\%$ ($p<0.05$) at 25 μM.

### 3.5 4NQO-induced DNA and mitochondrial damages independent of ROS production

Flow cytometric assessment of intracellular ROS namely superoxide and hydrogen peroxide levels using HE and DCFH-DA staining was performed to determine the role of ROS in 4NQO-induced DNA and mitochondrial damages. Interestingly, as shown in Fig 6A and 6B, there were no inductions of superoxide and hydrogen peroxide levels in 4NQO-treated cells as compared to control cells. Hydroquinone (HQ), which was used as positive control, had significantly increased ROS level in CCD-18Co cells ($p<0.05$). This suggested that ROS was not involved in DNA and mitochondrial damages caused by 4NQO, also leading to indication that neither cytoplasm nor mitochondria played a role in ROS production in 4NQO-treated cells. In addition, BK3C231 demonstrated potential as antioxidant by inhibiting HQ-induced ROS production.

### 3.6 Inhibition of 4NQO-induced nitrosative stress by BK3C231

Intracellular nitric oxide (NO) level was assessed using BD Pharmingen™ Orange NO Probe staining whereas extracellular NO level was assessed using Griess assay to determine the involvement of RNS in 4NQO-induced DNA and mitochondrial damages. Our study demonstrated a significant 0.98-fold increase of intracellular NO level, $15.7 \pm 0.19\%$ and 2.4-fold increase of extracellular NO level, $5.15 \pm 0.17$ μM ($p<0.05$) in 4NQO-treated cells over control cells, at $7.63 \pm 0.19\%$ and $1.51 \pm 0.26$ μM respectively, thereby demonstrating the involvement of NO in 4NQO-induced DNA and mitochondrial damages. Moreover, both BK3C231 and positive control, resveratrol significantly inhibited 4NQO-induced NO production from as early as 2 hours up till 24 hours of pretreatment ($p<0.05$) and were able to restore NO level back to basal level at 24 hours of pretreatment (Fig 7A and 7B).

In addition to that, antioxidant GSH level was assessed using Ellman's reagent. 4NQO-treated cells showed a reduced GSH level at $194.70 \pm 23.83$ nmol/mg as compared to untreated control cells at $245.96 \pm 12.44$ nmol/mg (Fig 7C). Overall, the simultaneous increase in NO level and decrease in GSH level by 4NQO further confirmed the involvement of nitrosative stress in 4NQO-induced DNA and mitochondrial damages. However, no induction of GSH

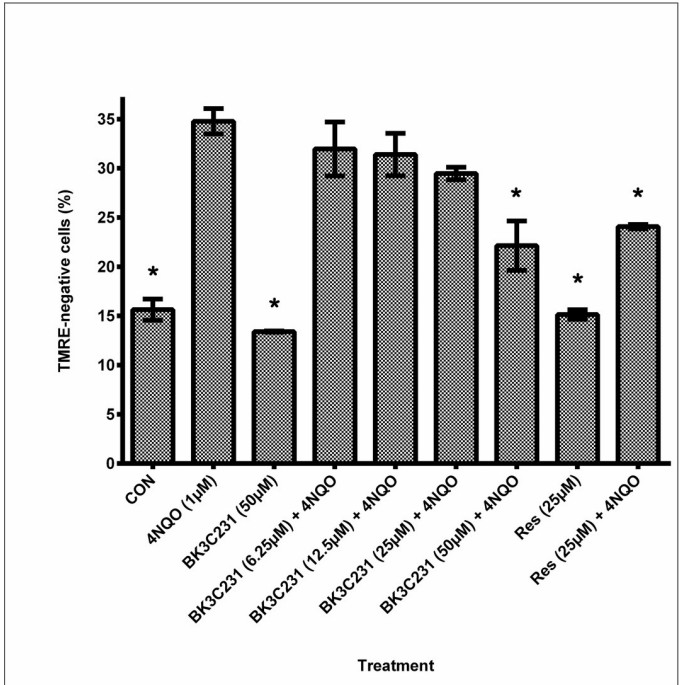 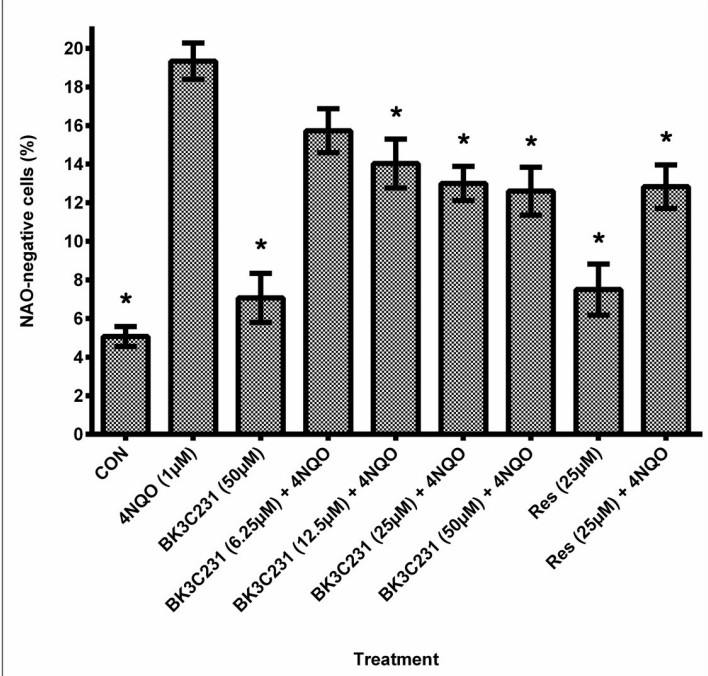

**Fig 5. Assessment of mitochondrial toxicity in CCD-18Co cells.** (A) Flow cytometric analysis of ΔΨm level using TMRE staining. (B) Flow cytometric analysis of cardiolipin level using NAO staining. Cells were pretreated with BK3C231 from 6.25 μM till 50 μM and resveratrol (res) at 25 μM for 2h prior to 4NQO induction at 1 μM for 2h. Each data point was obtained from three independent experimental replicates and expressed as mean ± SEM of TMRE- or NAO-negative cells (%). $^*$ p<0.05 against positive control, 4NQO only.

level was observed in cells pretreated with BK3C231 for 2 hours, 4 hours, 6 hours and 12 hours. BK3C231 was only able to significantly increase GSH level, 313.97 ± 27.83 nmol/mg (p<0.05) at 24 hours of pretreatment as compared to that of 4NQO-treated cells (Fig 7C). This

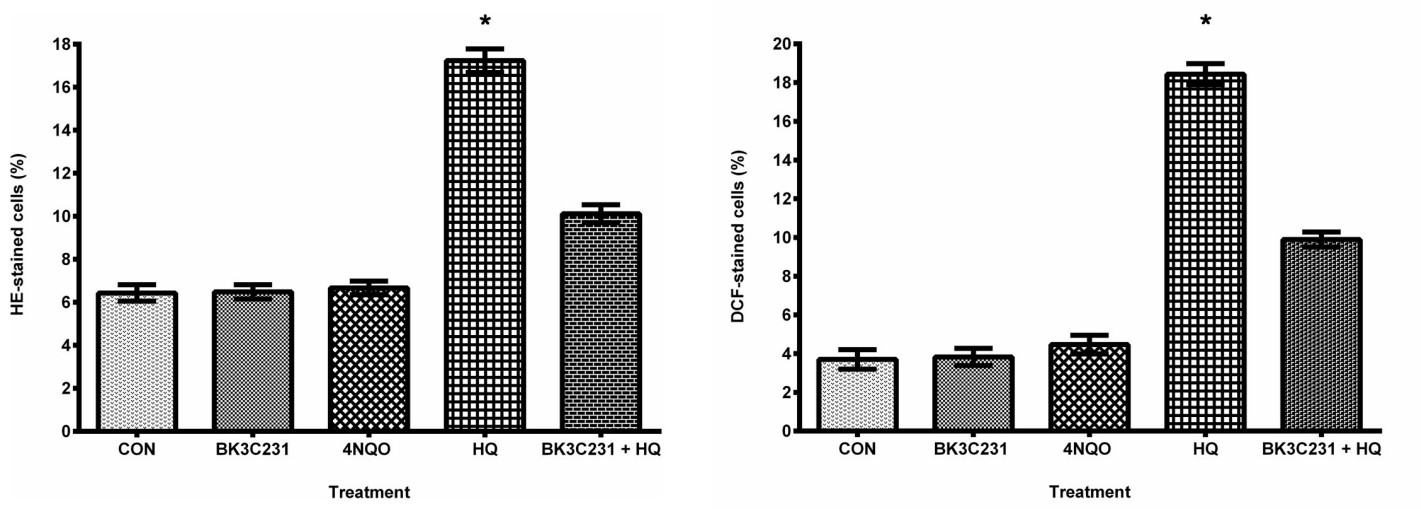

**Fig 6. Assessment of ROS production in CCD-18Co cells.** (A) Flow cytometric analysis of superoxide level using HE staining. (B) Flow cytometric analysis of hydrogen peroxide level using DCFH-DA staining. Cells were treated with BK3C231 at 50 μM for 24h and 4NQO at 1 μM for 1h. HQ treatment at 50 μM for 2h was used as positive control in this assay. Cells were also pretreated with BK3C231 at 50 μM for 24h prior to HQ induction at 50 μM for 2h. Each data point was obtained from three independent experimental replicates and expressed as mean ± SEM of HE- or DCF-stained cells (%). $^*$ p<0.05 against negative control, CON.

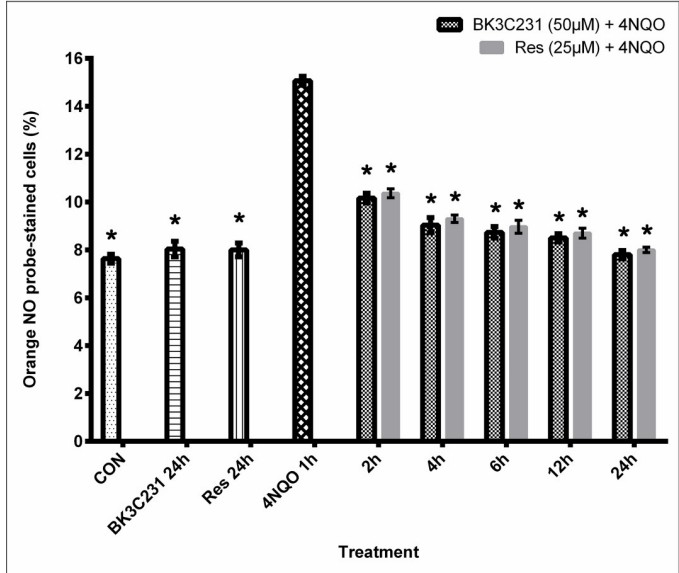

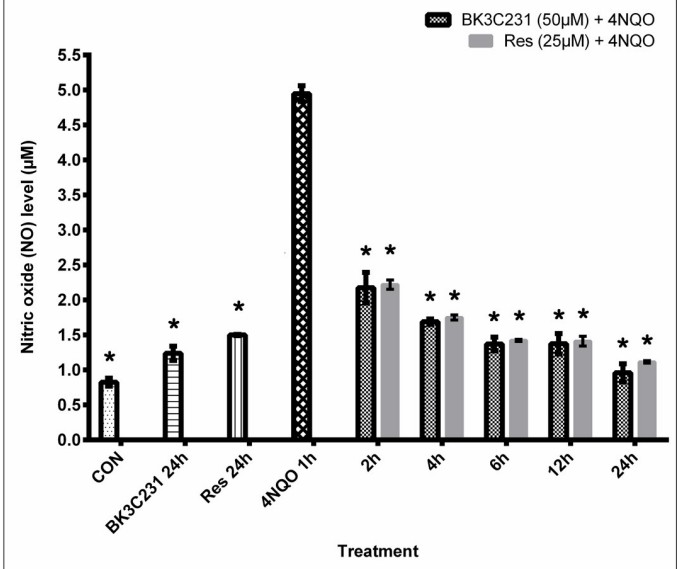

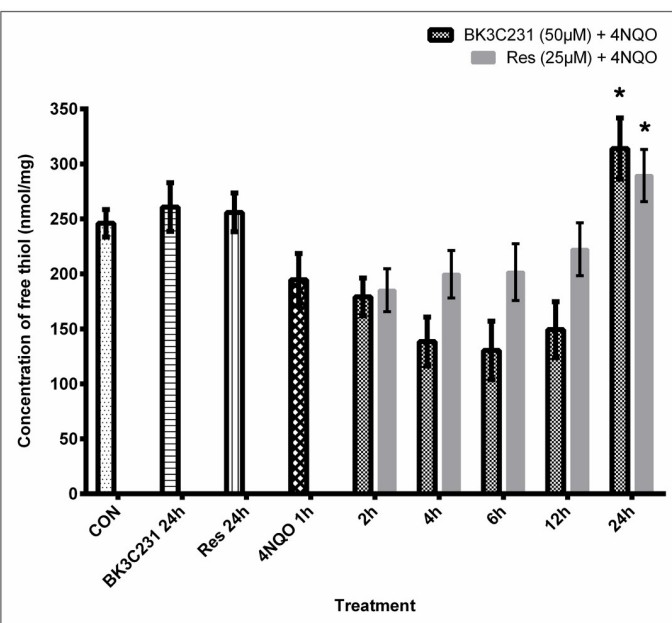

**Fig 7. Nitrosative stress assessment in CCD-18Co cells.** (A) Determination of intracellular NO level using Orange NO probe staining. (B) Determination of extracellular NO level using Griess assay. (C) Measurement of intracellular GSH level using Ellman's reagent. Cells were pretreated with BK3C231 at 50 μM and resveratrol (res) at 25 μM for 2h, 4h, 6h, 12h and 24h prior to 4NQO induction at 1 μM for 1h. Each data point was obtained from three independent experimental replicates and expressed as mean ± SEM of orange NO probe-stained cells (%), NO level and concentration of free thiol. * $p < 0.05$ against positive control, 4NQO only.

suggested that BK3C231 inhibited 4NQO-induced nitrosative stress through early reduction of NO production and late induction of GSH level in CCD-18Co cells.

## 4. Discussion

Epidemiological studies have shown that consumption of fruits particularly rich in stilbenes led to a reduced risk of colorectal cancer, which is one of the most commonly diagnosed cancers worldwide [40,41]. Furthermore, cytoprotection of DNA and mitochondrial function

limits the occurrence of cancer. Since DNA is the repository of hereditary material and genetic information in every living cell, the maintenance of its stability is pivotal as unrepaired DNA damage caused by diverse assaults from the environment, nutrition and natural cellular processes lead to cancer [42,43]. As for mitochondria, impairments and alterations of mitochondrial structure and functions, including morphology and redox potential, are associated with cancer transformation and have been frequently reported in human cancers [44–46]. In agreement with this, our study showed that BK3C231 was able to inhibit 4NQO-induced cytotoxicity as well as protect against DNA damage and mitochondrial dysfunction in the normal human colon fibroblast CCD-18Co cell line.

Firstly, we sought to understand the carcinogenic actions of 4NQO. Studies have demonstrated that 4NQO elicited carcinogenicity through its proximate carcinogenic metabolite namely 4-hydroxyaminoquinoline 1-oxide (4HAQO) produced by the enzymatic four-electron reduction of 4NQO's nitro group [47,48]. Being a potent chemical carcinogen and as a UV-mimetic agent, 4NQO is often used as positive control in various genotoxicity studies due to its well characterized metabolic processes [49]. As a result of a study by Brüsehafer et al. [50], it was reported that 4NQO predominantly induces mutagenicity more than clastogenicity and that the latter depends on cell types, our study has proved that 4NQO is a significant cause of DNA damage via DNA strand breaks and chromosomal damage via micronucleus formation. Our study was also in agreement with previous studies which demonstrated that 4NQO caused damage to mitochondrial membrane as characterized by loss of mitochondrial membrane potential ($\Delta\Psi$m) and cardiolipin [51].

As 4HAQO's carcinogenic effect is mainly based on DNA adduct formation, our study investigated 4NQO's other carcinogenic mechanism of action through generation of ROS and RNS and its involvement in the cytoprotective role of BK3C231 [52–54]. Interestingly, our study which revealed no superoxide and hydrogen peroxide production by 4NQO at 1 μM for 1 hour in CCD-18Co cells contradicts the study by Arima et al. [37] who reported ROS formation in human primary skin fibroblast by 4NQO using the same treatment concentration and timepoint. The discrepancy is likely due to the difference in the origin of fibroblast used. Hence, our study is the first to elucidate such findings on 4NQO mechanism which has never been shown in other studies thus far.

In addition, our study demonstrated an increased NO level and a depleted GSH level by 4NQO. This is possibly due to formation of 4NQO-GSH conjugates leading to generation of nitrite, a stable end product of NO, which inactivated γ-glutamylcysteine synthase and therefore suppressed intracellular synthesis of GSH [37,54–56]. Our data was also in agreement with previous studies that NO could be the main culprit in 4NQO-induced DNA and mitochondrial damages in CCD-18Co cells as NO has been demonstrated to induce genotoxicity and damage to mitochondria via multiple mechanisms directly or indirectly [57,58]. Moreover, the concurrent increase in NO level and decrease in GSH level postulates the occurrence of nitrosative stress which may contribute to 4NQO-induced DNA and mitochondrial damage [59,60].

More importantly, BK3C231 was shown in our study to protect against 4NQO-induced DNA and mitochondrial damages by decreasing DNA strand breaks and micronucleus formation as well as reducing loss of mitochondrial membrane potential ($\Delta\Psi$m) and cardiolipin (Fig 8). Our study further revealed that BK3C231 exerted these cytoprotective effects in CCD-18Co cells by suppressing 4NQO-induced nitrosative stress through reduction in NO level and late upregulation of GSH level. Furthermore, inhibition of HQ-induced ROS production by BK3C231 as demonstrated in this study corroborates the potential of BK3C231 as an antioxidant. The role of stilbene derivatives as potential antioxidants has been demonstrated in several studies involving for example resveratrol, a well-known stilbenoid, which attenuates

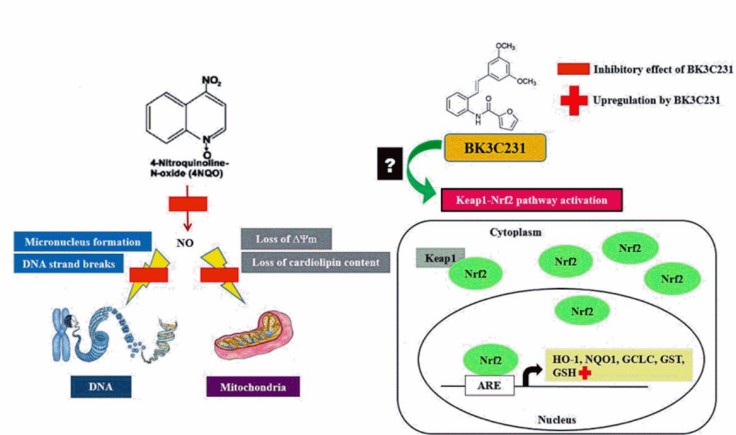

**Fig 8. Schematic representation of BK3C231-induced cytoprotection against 4NQO damage in CCD-18Co human colon fibroblast cells.** 4NQO caused DNA strand breaks and micronucleus formation as well as mitochondrial membrane potential ($\Delta\Psi$m) and cardiolipin losses in CCD-18Co cells through NO formation. BK3C231 inhibited these 4NQO-induced DNA and mitochondrial damages by decreasing NO level and increasing GSH level.

nitrosative stress in the small intestine of rats [61]. Piceatannol and isorhapontigenin, which are natural occurring stilbenes, have also been demonstrated to scavenge NO and nitrogen dioxide ($NO_2$) radicals as well as increasing GSH/GSSG ratio [62,63]. Therefore, we have included resveratrol, which is under the same stilbene family as BK3C231, as the positive control in this study which supported the BK3C231 findings and were consistent with previous studies which had reported the protective effects of resveratrol on mitochondrial function as well as its role in NO suppression and GSH induction [64–68].

Nitrosative stress is one of the most critical factors in multi-stage carcinogenesis with NO playing an important role in tumour biology and overproduction of NO can promote tumour growth [69]. Kee et al. [36] has reported the chemopreventive activity of BK3C231 involving upregulation of the detoxifying enzyme NQO1 due to the presence of methoxy and furan carboxamide groups. Therefore, it is possible that the presence and substitution pattern in relation to the methoxy group enables BK3C231 to act as an NO scavenger.

The Keap1-Nrf2 signaling pathway may also be involved in the depletion of NO level by BK3C231. The upstream Keap1-Nrf2 signaling pathway, which is a major regulator of phase II detoxification and cytoprotective genes, is postulated to be involved through upregulation of detoxifying enzymes which may lead to NO suppression [70]. Stilbene derivatives particularly resveratrol play a significant role in the activation of Nrf2-related gene transcription which induces expression of cytoprotective enzymes such as NQO1, glutathione S-transferase (GST), glutamate-cysteine ligase catalytic subunit (GCLC) and heme oxygenase-1 (HO-1) thus leading to protection against cancer [71]. Moreover, the increase in GSH as observed in this study can be attributed to the redox-sensitive Nrf2 activation [72,73].

Cytoprotection plays an important role in chemoprevention by suppressing the initiation stage of carcinogenesis. Cytoprotective mechanisms in response to DNA and mitochondrial damages by key causes of malignant transformation such as electrophile, oxidative stress and nitrosative stress represent a target for chemopreventive agent [74]. Our study has demonstrated the cytoprotective role of BK3C231, therefore it warrants further investigation of the chemopreventive role of BK3C231 in the Keap1-Nrf2 pathway which serves as both an important target thus provide significant protection of normal cells against carcinogenesis.

## 5. Conclusion

In conclusion, this study has provided a good insight into 4NQO-induced carcinogenicity in CCD-18Co cells. The demonstration of BK3C231 as a cytoprotective agent also served as a stepping stone for further elucidation of its chemopreventive potential against both genetic and epigenetic bases of cancer development. Based on our current findings in this study, we aim to reduce the gap between understanding molecular mechanism occurring in cancer carcinogenesis and instigating successful adoption of chemoprevention using BK3C231.

## Acknowledgments

The authors would like to thank Dr. Kee Chin Hui, previously of the Department of Chemistry, Faculty of Science, University of Malaya for her contribution in the synthesis of compound BK3C231.

## Author Contributions

**Conceptualization:** Kok Meng Chan.

**Data curation:** Huan Huan Tan.

**Formal analysis:** Huan Huan Tan, Kok Meng Chan.

**Funding acquisition:** Kok Meng Chan.

**Investigation:** Huan Huan Tan, Kok Meng Chan.

**Methodology:** Huan Huan Tan, Kok Meng Chan.

**Project administration:** Huan Huan Tan, Kok Meng Chan.

**Resources:** Noel Francis Thomas, Salmaan Hussain Inayat-Hussain, Kok Meng Chan.

**Supervision:** Noel Francis Thomas, Salmaan Hussain Inayat-Hussain, Kok Meng Chan.

**Validation:** Huan Huan Tan, Kok Meng Chan.

**Visualization:** Huan Huan Tan, Kok Meng Chan.

**Writing – original draft:** Huan Huan Tan.

**Writing – review & editing:** Huan Huan Tan, Noel Francis Thomas, Salmaan Hussain Inayat-Hussain, Kok Meng Chan.

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
