## [Decision Letter · Decision Letter 0]

30 Jan 2020

PONE-D-19-26183

Cytoprotective effects of (E)-N-(2-(3, 5-dimethoxystyryl) phenyl) furan-2-carboxamide (BK3C231) against 4-nitroquinoline 1-oxide-induced damage in CCD-18Co human colon fibroblast cells

PLOS ONE

Dear Dr. Chan,

Thank you for submitting your manuscript to PLOS ONE. After careful consideration, we feel that it has merit but does not fully meet PLOS ONE’s publication criteria as it currently stands. Therefore, we invite you to submit a revised version of the manuscript that addresses the points raised during the review process.

Please perform additional experiments to prove your conclusions as requested by reviewer 1.

Figures have to be modified, and the manuscript should be carefully edited by an English native speaker.

Please clarify the link between chemopreventive and cytoprotective effects of BK3C231

We would appreciate receiving your revised manuscript by Mar 15 2020 11:59PM. To enhance the reproducibility of your results, we recommend that if applicable you deposit your laboratory protocols in protocols.io, where a protocol can be assigned its own identifier (DOI) such that it can be cited independently in the future. For instructions see: http://journals.plos.org/plosone/s/submission-guidelines#loc-laboratory-protocols

We look forward to receiving your revised manuscript.

Kind regards,

Irina V. Lebedeva, Ph.D.

Academic Editor

PLOS ONE

Journal Requirements:

3. Please ensure that you refer to Figure 10 in your text as, if accepted, production will need this reference to link the reader to the figure.

Reviewers' comments:

Reviewer's Responses to Questions

**Comments to the Author**

1. Is the manuscript technically sound, and do the data support the conclusions?

Reviewer #1: Partly

Reviewer #2: Partly

2. Has the statistical analysis been performed appropriately and rigorously? 

Reviewer #1: Yes

Reviewer #2: Yes

3. Have the authors made all data underlying the findings in their manuscript fully available?

Reviewer #1: Yes

Reviewer #2: Yes

4. Is the manuscript presented in an intelligible fashion and written in standard English?

Reviewer #1: No

Reviewer #2: Yes

5. Review Comments to the Author

Reviewer #1: In the manuscript titled “Cytoprotective effects of (E)-N-(2-(3, 5-dimethoxystyryl) phenyl) furan-2-carboxamide (BK3C231) against 4-nitroquinoline 1-oxide-induced damage in CCD-18Co human colon fibroblast cells” Tan et al. investigate the cytoprotective role of BK3C231 on 4NQO induced damages on DNA and mitochondria were investigated in normal human colon fibroblast, CCD-18Co cells.

Authors demonstrate that BK3C231 protected normal human colon fibroblast, CCD-18Co cells. against DNA strand breaks, micronucleus formation as well as dissipation of mitochondrial membrane potential (ΔΨm) and cardiolipin induced by 4 NQO treatment. These effects seem to be ascribed to the ability of BK3C231 in causing NO downregulation as well as an increase in GSH level.

The manuscript is interesting, however authors need to provide additional clues of the mechanism described.

Major point

• Authors exclude that ROS production can play a role in 4NQO mechanism that seems, indeed, due to RNS generation as demonstrated by Griess test. To better clarify this aspect authors should report data of both ΔΨm and NO production in the presence of a positive control as a NO inhibitor, such as L-NAME.

• A more specific analysis of nitrosative stress as a consequence of an oxidative unbalance should be confirmed by an analysis focused on intracellular content of thiol groups as well as carbonylated proteins.

• To exclude the production of oxidative stress from mitochondrial environment author should explore a possible anion superoxide measuring by the fluorogenic probe MitoSox Red, also in the presence of antimycin A.

Minor points

The manuscript should carefully read a mother tongue English speaker.

Reviewer #2: Comments to the Authors

The manuscript from Huan Huan Tan et al. entitled “Cytoprotective effects of (E)-N-(2-(3, 5-dimethoxystyryl) phenyl) furan-2-carboxamide (BK3C231) against 4-nitroquinoline 1-oxide-induced damage in CCD-18Co human colon fibroblast cells” (Manuscript Number: PONE-D-19-26183) seems to be an interesting manuscript. There are points which need to be addressed.

1. There seems so many figures and each figure has only one or two graphs. Reviewers recommend authors to unite them.

2. The horizontal axis in some figures are not easy to understand; for example Figs. 4A, 4B, 5B, 6, 7, 9A, 9B, and 9C. Authors can improve them so that readers understand; for example using plus (+) and minus (-) signs.

3. Reviewers do not think the findings in this study indicate chemoprevnetive potential of BK3C231, but only cytoprotective potential. Several parts of manuscript should be changed.

6. PLOS authors have the option to publish the peer review history of their article (what does this mean?). If published, this will include your full peer review and any attached files.

Reviewer #1: No

Reviewer #2: No

---

## [Author Response · Author response to Decision Letter 0]

23 Mar 2020

Dear Dr. Lebedeva,

We would like to thank you and the reviewers for your insightful and constructive comments to help improve our manuscript entitled “Cytoprotective effects of (E)-N-(2-(3, 5-dimethoxystyryl) phenyl) furan-2-carboxamide (BK3C231) against 4-nitroquinoline 1-oxide-induced damage in CCD18-Co human colon fibroblast cells”.

Please kindly find our detailed responses to all the comments raised as below.

Author response to academic editor 

Response to editor’s comment no.1: Manuscript has been checked thoroughly to meet PLOS ONE's style requirements.

Response to editor’s comment no. 2: The phrase “data not shown” has been removed and we have included a table with reference to the text. 

3. Please ensure that you refer to Figure 10 in your text as, if accepted, production will need this reference to link the reader to the figure.

Response to editor’s comment no. 3: Figure 10 has been changed to Figure 8 and the figure has been referred to in the revised manuscript’s text.

Author response to reviewers

Reviewer #1: In the manuscript titled “Cytoprotective effects of (E)-N-(2-(3, 5-dimethoxystyryl) phenyl) furan-2-carboxamide (BK3C231) against 4-nitroquinoline 1-oxide-induced damage in CCD-18Co human colon fibroblast cells” Tan et al. investigate the cytoprotective role of BK3C231 on 4NQO induced damages on DNA and mitochondria were investigated in normal human colon fibroblast, CCD-18Co cells.

Authors demonstrate that BK3C231 protected normal human colon fibroblast, CCD-18Co cells. against DNA strand breaks, micronucleus formation as well as dissipation of mitochondrial membrane potential (ΔΨm) and cardiolipin induced by 4NQO treatment. These effects seem to be ascribed to the ability of BK3C231 in causing NO downregulation as well as an increase in GSH level.

The manuscript is interesting, however authors need to provide additional clues of the mechanism described.

Major point

• Authors exclude that ROS production can play a role in 4NQO mechanism that seems, indeed, due to RNS generation as demonstrated by Griess test. To better clarify this aspect authors should report data of both ΔΨm and NO production in the presence of a positive control as a NO inhibitor, such as L-NAME.

Response to reviewer #1’s comment: We thank the reviewer for the suggestion. We have included resveratrol as positive control instead of L-NAME to support our findings in the revised manuscript. We use resveratrol as a positive control because resveratrol falls under the same stilbene family as BK3C231 and has been shown by several studies to protect against ΔΨm loss and inhibit NO generation (Chen et al. 2017, doi: 10.1016/j.physbeh.2017.09.024; Xu et al. 2016, doi: 10.1038/nchembio.2113; Kimbrough et al. 2015, doi: 10.1016/j.surg.2015.07.012). In this study, RV indeed protected 4NQO-treated cells against ΔΨm loss and suppressed NO generation.

• A more specific analysis of nitrosative stress as a consequence of an oxidative unbalance should be confirmed by an analysis focused on intracellular content of thiol groups as well as carbonylated proteins.

Response to reviewer #1’s comment: We have already included analysis of intracellular content of thiol groups namely glutathione (GSH) using Ellman’s reagent (Section 3.6, Figure 7C). This method is a sensitive and accurate measurement of intracellular GSH (Rahman et al. 2006, doi: 10.1038/nprot.2006.378) which contributes to specific analysis of nitrosative stress in addition to the assessment of intracellular and extracellular NO level in our study.

• To exclude the production of oxidative stress from mitochondrial environment author should explore a possible anion superoxide measuring by the fluorogenic probe MitoSox Red, also in the presence of antimycin A.

Response to reviewer #1’s comment: We acknowledged the reviewer’s suggestion to use MitoSox Red to measure mitochondrial superoxide anion, however, in this study, we have included flow cytometric assessment of superoxide anion level using HE staining and hydrogen peroxide level using DCFH-DA staining, with the inclusion of positive control for ROS generation which is hydroquinone (HQ). Both HE and DCFH-DA are ideal, widely-used probes to detect intracellular oxidant formation and they serve as sensitive, specific and direct techniques to measure the redox state of a cell regardless of the origin of ROS, whether in cytoplasm or mitochondria (Kalyanaraman et al. 2012, doi: 10.1016/j.freeradbiomed.2011.09.030; Eruslanov & Kusmartsev 2010, doi: 10.1007/978-1-60761-411-1_4). As discussed in Section 4, 4NQO is able to generate ROS directly without involvement of mitochondria, even in a cell-free system (Arima et al. 2006, doi: 10.1093/toxsci/kfj161). This prompted us to determine the role of ROS in 4NQO-induced genotoxicity and mitochondrial toxicity prior to elucidating the cytoprotective role of BK3C231. Through these assessments, our study demonstrated that there were no inductions of intracellular superoxide and hydrogen peroxide levels in 4NQO-treated cells as shown in Figure 6, leading to indication that neither cytoplasm nor mitochondria played a role in ROS production in 4NQO-treated cells. Interestingly, BK3C231 demonstrated potential as antioxidant by inhibiting HQ-induced ROS production. Therefore, we deemed these findings sufficient to fulfil one of the objectives in our study.

Minor points

The manuscript should carefully read a mother tongue English speaker.

Response to reviewer #1’s comment: The revised manuscript has been carefully edited by an English native speaker. One of our authors, Dr. Noel Thomas is an English native speaker from the United Kingdom who is currently residing in Malaysia.

Reviewer #2: Comments to the Authors

The manuscript from Huan Huan Tan et al. entitled “Cytoprotective effects of (E)-N-(2-(3, 5-dimethoxystyryl) phenyl) furan-2-carboxamide (BK3C231) against 4-nitroquinoline 1-oxide-induced damage in CCD-18Co human colon fibroblast cells” (Manuscript Number: PONE-D-19-26183) seems to be an interesting manuscript. There are points which need to be addressed.

1. There seems so many figures and each figure has only one or two graphs. Reviewers recommend authors to unite them.

Response to reviewer #2’s comment: We thank the reviewer for the recommendation. Some of the figures have been united in the revised manuscript.

2. The horizontal axis in some figures are not easy to understand; for example Figs. 4A, 4B, 5B, 6, 7, 9A, 9B, and 9C. Authors can improve them so that readers understand; for example using plus (+) and minus (-) signs.

Response to reviewer #2’s comment: The horizontal axis especially in the mentioned figures have been revised for better understanding to readers.

3. Reviewers do not think the findings in this study indicate chemopreventive potential of BK3C231, but only cytoprotective potential. Several parts of manuscript should be changed.

Response to reviewer #2’s comment: We agree with the reviewer and certain parts of the manuscript have been changed and rephrased to indicate BK3C231’s cytoprotective potential. Cytoprotection contributes as one of the steps in chemoprevention by suppressing the initiation stage of carcinogenesis caused by carcinogen-induced cellular damages on DNA and mitochondria. This study has demonstrated the cytoprotective role of BK3C231 and therefore warrants further investigation on BK3C231’s chemopreventive potential in the near future. 

We hope that you find the paper acceptable for publication in PLOS ONE. We look forward to hearing from you soon.

Thank you.

Kind regards,

Chan Kok Meng

---

## [Decision Letter · Decision Letter 1]

16 Apr 2020

Cytoprotective effects of (E)-N-(2-(3, 5-dimethoxystyryl) phenyl) furan-2-carboxamide (BK3C231) against 4-nitroquinoline 1-oxide-induced damage in CCD-18Co human colon fibroblast cells

PONE-D-19-26183R1

Dear Dr. Chan,

We are pleased to inform you that your manuscript has been judged scientifically suitable for publication and will be formally accepted for publication once it complies with all outstanding technical requirements.

With kind regards,

Irina V. Lebedeva, Ph.D.

Academic Editor

PLOS ONE

Additional Editor Comments (optional):

Reviewers' comments:

Reviewer's Responses to Questions

**Comments to the Author**

1. If the authors have adequately addressed your comments raised in a previous round of review and you feel that this manuscript is now acceptable for publication, you may indicate that here to bypass the “Comments to the Author” section, enter your conflict of interest statement in the “Confidential to Editor” section, and submit your "Accept" recommendation.

Reviewer #1: All comments have been addressed

Reviewer #2: All comments have been addressed

2. Is the manuscript technically sound, and do the data support the conclusions?

Reviewer #1: Yes

Reviewer #2: Yes

3. Has the statistical analysis been performed appropriately and rigorously? 

Reviewer #1: Yes

Reviewer #2: Yes

4. Have the authors made all data underlying the findings in their manuscript fully available?

Reviewer #1: Yes

Reviewer #2: Yes

5. Is the manuscript presented in an intelligible fashion and written in standard English?

Reviewer #1: Yes

Reviewer #2: Yes

6. Review Comments to the Author

Reviewer #1: (No Response)

Reviewer #2: Comments to the Authors

According to the comments from the Reviewers, the Authors responded adequately and conducted several modifications appropriately. This seems a quite well-written and reshaped manuscript. Therefore, this can be suitable for publication in the journal.

7. PLOS authors have the option to publish the peer review history of their article (what does this mean?). If published, this will include your full peer review and any attached files.

Reviewer #1: No

Reviewer #2: No

---

## [Editor Report · Acceptance letter]

22 Apr 2020

PONE-D-19-26183R1 

Cytoprotective effects of (E)-N-(2-(3, 5-dimethoxystyryl) phenyl) furan-2-carboxamide (BK3C231) against 4-nitroquinoline 1-oxide-induced damage in CCD-18Co human colon fibroblast cells 

Dear Dr. Chan:

I am pleased to inform you that your manuscript has been deemed suitable for publication in PLOS ONE. Congratulations! Your manuscript is now with our production department. 

With kind regards,

on behalf of

Dr. Irina V. Lebedeva 

Academic Editor

PLOS ONE